# Dynamic Behavior of a Predator–Prey Model with Double Delays and Beddington–DeAngelis Functional Response

Minjuan Cui [1], Yuanfu Shao [1,*], Renxiu Xue [1] and Jinxing Zhao [2]

1   College of Science, Guilin University of Technology, Guilin 541004, China
2   School of Mathematical Sciences, Inner Mongolia University, Hohhot 010021, China
*   Correspondence: shaoyuanfu@163.com

**Abstract:** In the predator–prey system, predators can affect the prey population by direct killing and predation fear. In the present study, we consider a delayed predator–prey model with fear and Beddington–DeAngelis functional response. The model incorporates not only the fear of predator on prey with an intraspecific competition relationship, but also fear delay and pregnancy delay. Apart from the local stability analysis of the equilibrium points of the model, we find that time delay can change the stability of the system and cause Hopf bifurcation. Taking time delay as the bifurcation parameter, the critical values of delays in several cases are derived. In addition, we extend it to the random environment and study the stochastic ultimate boundedness of the stochastic process. Finally, our theoretical results are validated by numerical simulation.

**Keywords:** delay; equilibrium state; stability; Hopf bifurcation

**MSC:** 34C23; 60H10; 92D25





## 1. Introduction

Predator–prey interaction is an important relationship in population dynamics, which is a hot research topic in ecology and biomathematics. The study of predator–prey models can help us prevent and control large-scale species changes, protect species diversity, and optimize ecosystems.

An important component of the predator–prey relationship is the functional response of the predator. The functional response refers to the effect of the amount of predation by a predator in a certain period of time on the change of prey density. Numerous experiments [1,2] have shown that the functional response depends not only on the density of prey but also on the density of predators. Pal et al. [3] studied the predator–prey model using the Beddington–DeAngelis functional response, and they proved the asymptotic dynamic properties and explored the influence of fear on the stability of the system. Huang et al. [4] considered a delayed virus dynamics model with the Beddington–DeAngelis functional response and studied its stability analysis.

For a long time, predator–prey systems were studied only in terms of the effect of direct killing by predators [5,6]. However, experiments with song sparrows [7] and elk populations [8–11] have shown that fear of predators also affects prey populations, and sometimes the influence was even more serious than direct predation [12,13]. In addition, most studies have considered that fear reduces the reproductive rate of prey, whereas intraspecies competition is not affected [14–16]. In real life, fear may also have an effect on intraspecific competition for prey. Therefore, it is natural to include the fear of predator in our research model.

In fact, there exists time delay [17] in all kinds of biological processes, such as digestion of food, conversion of energy, gestation, maturation, inducible defense of prey groups, and so on. When prey senses a chemical signal or a sound signal, they need a certain

amount of time to react, known as the fear delay. There is also a time lag between eating the prey and producing offspring, known as delayed gestation. However, few studies have investigated both fear delay and pregnancy delay [15,18–20]. Moreover, time delay has an important effect on the stability of the system [21–23]. Panday et al. [24] mainly studied the local stability of the system and the direction and stability of Hopf bifurcation. Furthermore, they found that when the time delay exceeded the critical value, the system developed Hopf bifurcation and became unstable. Kumar and Dubey [25] investigated a prey–predator system with prey refuge and gestation delay, and they proved the global asymptotic stability of the model and investigated the Hopf bifurcation behavior induced by the fear effect of prey.

Based on the above analysis, we propose a predator–prey model with double delays with fear and Beddington–DeAngelis functional response:

$$\begin{cases} \frac{dx}{dt} = \frac{kx}{1+fy(t-\tau_1)} - \alpha x^2 - \frac{pxy}{ax+by+c}, \\ \frac{dy}{dt} = \frac{\mu px(t-\tau_2)y(t-\tau_2)}{ax(t-\tau_2)+by(t-\tau_2)+c} - dy - hy^2, \end{cases} \tag{1}$$

with the initial conditions

$$x(\theta) = \varphi_1(\theta), y(\theta) = \varphi_2(\theta), \varphi(\theta) = (\varphi_1(\theta), \varphi_2(\theta)) \in C([-\tau, 0], R_+^2), \tag{2}$$

where $\tau_1$ is the fear delay and $\tau_2$ is the gestation delay, $\tau = \max\{\tau_1, \tau_2\}$, $R_+^2 = \{(x, y) : x > 0, y > 0\}$, $k$ denotes the intrinsic growth rate of prey species, $f$ is the level of fear caused by predators, we denotes $p$ by the per capita predator consumption rate, $\alpha$ is the decay rate of prey due to intraspecies competition, $a$ is the time of the predator for each prey that is consumed, $b$ measures the mutual interference between predators, $c$ is the half-saturation constant of the predator population, $\mu$ is the conversion rate of prey biomass to predator, $d$ is the natural mortality rate of predators, and $h$ represented the death rate of predators from intraspecific competition. We define that $||\varphi|| = \max\{|\varphi(\theta)| : \theta \in [-\tau, 0]\}$.

Because the parameters of model (1) are fixed and the influence of environmental noise is not considered, we incorporated random perturbation in the model (1) to investigate the impact of white noise on population dynamics. We establish the stochastic differential equation by perturbing the birth rate of prey population and natural death rate of predator population [26]. Then, we acquire the following stochastic model:

$$\begin{cases} dx = \left(\frac{kx}{1+fy(t-\tau_1)} - \alpha x^2 - \frac{pxy}{ax+by+c}\right)dt + \frac{\sigma_1 x}{1+fy(t-\tau_1)}dB_1(t), \\ dy = \left(\frac{\mu px(t-\tau_2)y(t-\tau_2)}{ax(t-\tau_2)+by(t-\tau_2)+c} - dy - hy^2\right)dt + \sigma_2 y dB_2(t), \end{cases} \tag{3}$$

where $B_1(t)$ and $B_2(t)$ are standard and mutually independent Brownian motions defined on a complete probability space $(\Omega, \mathcal{F}, \mathcal{P})$ with a filtration $\{\mathcal{F}_t\}_{t \geq 0}$ satisfying the usual conditions and $\sigma_i^2 (i = 1, 2)$ represents the intensity of the white noise.

The rest of this paper is organized as follows. In Section 2, we establish the positivity and boundedness of the solution of the delay model (1). In Section 3, we discuss the existence of equilibrium points, study the local stability of each equilibrium point and the Hopf bifurcation with different time delays. We consider the stochastic delay model (3) and prove the globally unique existence and stochastic ultimate boundedness of positive solutions in Section 4. We will perform some numerical simulations in Section 5 to verify our theoretical results. Finally, we draw conclusions based on our research findings and put forward some suggestions for future work in Section 6.

## 2. Positivity and Boundedness

To ensure that the model has a biological background, we study the positivity and boundedness of the delay Equation (1), that is, the solution is positive and invariant in the first quadrant and does not go beyond the given interval.

### 2.1. Positivity

**Theorem 1.** *Let $(x(t), y(t))$ be the solution of system (1) with the initial conditions (2). Then, $(x(t), y(t))$ remains positive for any time $t \geq 0$.*

**Proof.** System (1) is rewritten as a matrix

$$\dot{H} = G(H), \tag{4}$$

where $H = \begin{bmatrix} x \\ y \end{bmatrix}$, $G(H) = \begin{bmatrix} G_1(H) \\ G_2(H) \end{bmatrix} = \begin{bmatrix} \frac{kx}{1+fy(t-\tau_1)} - \alpha x^2 - \frac{pxy}{ax+by+c} \\ \frac{\mu px(t-\tau_2)y(t-\tau_2)}{ax(t-\tau_2)+by(t-\tau_2)+c} - dy - hy^2 \end{bmatrix}$ with initial condition

$$H(\theta) = (\varphi_1(\theta), \varphi_2(\theta)) \in C([-\tau, 0], R_+^2), \varphi_i(0) > 0, i = 1, 2. \tag{5}$$

We can easily check the system (4) whenever taking $H(\theta) \in R_+^2$ such that $x = y = 0$. Then, we obtain

$$G_i(H)|_{h_i = 0, H \in R_+^2} \geq 0$$

with $h_1(t) = x(t), h_2(t) = y(t)$. According to Lemma 4 in [27], we derive that every solution of (4) with the initial condition (5) is positive, that is to say, any solution of the system (1) belongs to the region $R_+^2$ and remains positive for any $t \geq 0$. □

### 2.2. Boundedness

**Theorem 2.** *All solutions of system (1) that start in $\mathbb{R}_+^2$ are bounded.*

**Proof.** Let $W(t) = x(t) + \frac{y(t+\tau_2)}{\mu}$. The time derivative of $W(t)$ along the solution of (1) is

$$\frac{dW(t)}{dt} = \frac{dx(t)}{dt} + \frac{1}{\mu}\frac{dy(t+\tau_2)}{dt}$$

$$= \frac{kx}{1+fy(t-\tau_1)} - \alpha x^2 - \frac{d}{\mu}y(t+\tau_2) - \frac{h}{\mu}(y(t+\tau_2))^2$$

$$\leq kx - \alpha x^2 - \frac{d}{\mu}y(t+\tau_2).$$

Choose a constant $\sigma$ such that $\sigma < d$. Then,

$$\frac{dW(t)}{dt} + \sigma W(t) \leq x(k + \sigma - \alpha x) - \frac{d-\sigma}{\mu}y(t+\tau_2)$$

$$\leq \frac{(k+\sigma)^2}{4\alpha} := M.$$

Applying differential inequality theory results, we obtain

$$0 \leq W(t) \leq \frac{M}{\sigma}(1 - \exp(-\sigma t)) + W(x(0), y(0))\exp(-\sigma t).$$

As $t \to \infty$, we have $0 < W(t) \leq \frac{M}{\sigma}$. Therefore, all the solutions of system (1) are confined in the region

$$Z = \{(x, y) \in \mathbb{R}_+^2 : 0 \leq W(t) \leq \frac{M}{\sigma}\}.$$

□

### 3. Stability Analysis

In this section, we mainly study the stability of equilibrium points and Hopf bifurcation around $E^*(x^*, y^*)$ of (1).

*3.1. Equilibrium Points and Existence Criterion*

The model (1) has three positive equilibrium points:
(1) the trivial equilibrium point $E_0(0,0)$;
(2) the free equilibrium point $E_1(\frac{k}{\alpha}, 0)$;
(3) the coexistence equilibrium point $E^*(x^*, y^*)$ satisfying the following equation:

$$\begin{cases} \frac{k}{1+fy} - \alpha x - \frac{py}{ax+by+c} = 0, \\ \frac{\mu px}{ax+by+c} - d - hy = 0, \end{cases} \tag{6}$$

By solving the above equation, we obtain $x^* = \frac{(by^*+c)(d+hy^*)}{\mu p - a(d+hy^*)}$. The interior equilibrium point exists only if $\mu p - a(d + hy^*) > 0$ and $y^*$ is the equation $Ay^4 + By^3 + Cy^2 + Dy + E = 0$, where

$A = a^2 h^2 f p + b^2 \alpha f h \mu p,$

$B = \mu p b \alpha (2cfh + bdf + bh) + a^2 h^2 p + 2ahfp(ad - \mu p),$

$C = \mu p b \alpha (bd + 2cdf + 2ch) + h\mu p(abk + c^2 \alpha f) + fp(ad - \mu p)^2 + 2ahp(ad - \mu p),$

$D = c\mu p(2bd\alpha + ahk + c\alpha h + cd\alpha f) + \mu p bk(ad - \mu p) + p(ad - \mu p)^2,$

$E = \mu p c^2 d\alpha + \mu p ck(ad - \mu p).$

By the Descartes rule of sign and $A > 0$, we find that the existence of at least one positive root when $E < 0$ is guaranteed, that is, $k(ad - \mu p) + cd\alpha < 0$. For more details, we refer the reader to [26].

*3.2. Local Stability Analysis and Hopf Bifurcation of Equilibria*

In this section, we perform the local stability on the dynamics of the model system (1) around the prescribed equilibrium points.

Using the transformations $X = x - x^*, Y = y - y^*$, the linearized form of system (1) is

$$\frac{dX(t)}{dt} = J_0 X(t) + J_1 X(t - \tau_1) + J_2 X(t - \tau_2), \tag{7}$$

where $X(t) = [x(t), y(t)]^T$,

$$J_0 = \begin{bmatrix} \frac{k}{1+fy^*} - 2\alpha x^* - \frac{py^*(by^*+c)}{(ax^*+by^*+c)^2} & \frac{-px^*(ax^*+c)}{(ax^*+by^*+c)^2} \\ 0 & -(d+2hy^*) \end{bmatrix}, J_1 = \begin{bmatrix} 0 & \frac{-kfx^*}{(1+fy^*)^2} \\ 0 & 0 \end{bmatrix}, J_2 = \begin{bmatrix} 0 & 0 \\ \frac{\mu py^*(by^*+c)}{(ax^*+by^*+c)^2} & \frac{\mu px^*(ax^*+c)}{(ax^*+by^*+c)^2} \end{bmatrix}.$$

(1) At $E_0$:

$$J_{E_0} = \begin{bmatrix} k & 0 \\ 0 & -d \end{bmatrix}.$$

The Jacobian matrix has the eigenvalues $k$ and $-d$, which shows that $E_0$ is always unstable. Therefore, we draw the following conclusion:

**Theorem 3.** *The trivial equilibrium point $E_0$ is always unstable.*

(2) At $E_1$:

$$J_{E_1} = \begin{bmatrix} -k & -\frac{pk}{ak+c\alpha} - \frac{fk^2}{\alpha}e^{-\lambda\tau_1} \\ 0 & -d + \frac{\mu pk}{ak+c\alpha}e^{-\lambda\tau_2} \end{bmatrix},$$

and the characteristic equation is

$$(-k - \lambda)\left(-d + \frac{\mu pk}{ak + c\alpha}e^{-\lambda\tau_2} - \lambda\right) = 0.$$

Hence, we can derive $\lambda_1 = -k$, $\lambda_2 = -d + \frac{\mu pk}{ak+c\alpha}e^{-\lambda\tau_2}$.

Case I: $\tau_2 = 0$, $\lambda_2 = -d + \frac{\mu pk}{ak+c\alpha}$, $E_1$ is locally asymptotically stable if $d > \frac{\mu pk}{ak+c\alpha}$.

Case II: $\tau_2 > 0$,

$$-d + \frac{\mu pk}{ak + c\alpha}e^{-\lambda\tau_2} - \lambda = 0. \tag{8}$$

Assume that $\lambda = i\xi(\xi > 0)$ is a root of (8). Then, we have

$$-d(ak + c\alpha) + \mu pk(cos\xi\tau_2 - isin\xi\tau_2) - i\xi(ak + c\alpha) = 0.$$

Then, we separate the real and imaginary parts as follows:

$$-d(ak + c\alpha) + \mu pkcos\xi\tau_2 = 0, \tag{9}$$

$$\mu pksin\xi\tau_2 + \xi(ak + c\alpha) = 0. \tag{10}$$

By squaring and adding (9) and (10), we have

$$(ak + c\alpha)^2(d^2 + \xi^2) = (\mu pk)^2.$$

Therefore, we obtain

$$\xi^2 = \left(\frac{\mu pk}{ak + c\alpha}\right)^2 - d^2.$$

If $\frac{\mu pk}{ak+c\alpha} - d > 0$, the above Equation (8) has a positive root, which implies that $E_1$ is unstable;

If $\frac{\mu pk}{ak+c\alpha} - d < 0$, Equation (8) contains one negative real root and imaginary roots with negative real parts, so $E_1$ is locally asymptotically stable for any $\tau_2 > 0$.

Therefore, we have the following conclusion:

**Theorem 4.** *The free equilibrium point $E_1$ of the system* (1) *is locally asymptotically stable for any $\tau_2 \geq 0$ if $\frac{\mu pk}{ak+c\alpha} - d < 0$.*

**Remark 1.** *$E_1$ is locally asymptotically stable if $ad + cd - \mu p > 0$, which contradicts the existence of $E^*$. In other words, the instability of $E_1$ guarantees the existence of $E^*$.*

(3) At $E^*$:

$$J_{E^*} = \begin{bmatrix} a_{11} & b_{12}e^{-\lambda\tau_1} + a_{12} \\ c_{21}e^{-\lambda\tau_2} & a_{22} + c_{22}e^{-\lambda\tau_2} \end{bmatrix},$$

where $a_{11} = \frac{k}{1+fy^*} - 2\alpha x^* - \frac{py^*(by^*+c)}{(ax^*+by^*+c)^2} = -\alpha x^* + \frac{apx^*y^*}{(ax^*+by^*+c)^2}$, $b_{12} = \frac{-fkx^*}{(1+fy^*)^2}$,

$a_{12} = -\frac{px^*(ax^*+c)}{(ax^*+by^*+c)^2}$, $c_{21} = \frac{\mu py^*(by^*+c)}{(ax^*+by^*+c)^2}$, $a_{22} = -(d + 2hy^*)$, $c_{22} = \frac{\mu px^*(ax^*+c)}{(ax^*+by^*+c)^2}$.

Characteristic equation at $E^*$ is given by

$$\lambda^2 + A_1\lambda + A_2\lambda e^{-\lambda\tau_2} + A_3e^{-\lambda\tau_2} + A_4 + A_5e^{-\lambda(\tau_1+\tau_2)} = 0, \tag{11}$$

where $A_1 = -(a_{11} + a_{22})$, $A_2 = -c_{22}$, $A_3 = a_{11}c_{22} - a_{12}c_{21}$, $A_4 = a_{11}a_{22}$, $A_5 = -b_{12}c_{21}$.

Case I: $\tau_1 = 0$, $\tau_2 = 0$.

The characteristic Equation (11) becomes

$$\lambda^2 + (A_1 + A_2)\lambda + A_3 + A_4 + A_5 = 0.$$

According to the Routh–Hurwitz criteria, we can infer the $J_{E^*}$ has a negative real part if $A_1 + A_2 > 0$, $A_3 + A_4 + A_5 > 0$, which implies $\frac{apy^*}{\alpha(ax^*+by^*+c)^2} < 1$. Hence, $E^*$ is locally asymptotically stable if $\frac{apy^*}{\alpha(ax^*+by^*+c)^2} < 1$.

Thus, we have the following theorem:

**Theorem 5.** *In the absence of both $\tau_1$ and $\tau_2$, the coexistence equilibrium point $E^*$ of the system (1) is locally asymptotically stable if $\frac{apy^*}{\alpha(ax^*+by^*+c)^2} < 1$.*

Case II: $\tau_1 > 0$, $\tau_2 = 0$. The characteristic Equation (11) becomes

$$\lambda^2 + (A_1 + A_2)\lambda + A_3 + A_4 + A_5 e^{-\lambda\tau_1} = 0. \tag{12}$$

Taking $\lambda = i\xi^*$. Then, we obtain the real and imaginary parts, respectively, as follows:

$$-(\xi^*)^2 + A_3 + A_4 + A_5 cos\xi^*\tau_1 = 0, \tag{13}$$

$$(A_1 + A_2)\xi^* - A_5 sin\xi^*\tau_1 = 0. \tag{14}$$

Squaring (13) and (14) and adding, we obtain

$$(\xi^*)^4 + R_1(\xi^*)^2 + R_2 = 0, \tag{15}$$

where $R_1 = (A_1 + A_2)^2 - 2(A_3 + A_4)$, $R_2 = (A_3 + A_4)^2 - A_5^2$. By calculating, we have

$$(\xi^*)^2 = \frac{-R_1 \pm \sqrt{R_1^2 - 4R_2}}{2}. \tag{16}$$

Let $Z(\xi^*) = (\xi^*)^4 + R_1(\xi^*)^2 + R_2$, then $Z(0) = R_2 < 0$, that is, $(A_3 + A_4)^2 < A_5^2$. Equation (16) has at least positive root $(\xi_0^*)^2$. So, Equation (12) has a pair of imaginary roots $\pm i\xi_0^*$. Substituting $(\xi_0^*)^2$ in (13) and (14), we obtain

$$\tau_{1_k} = \frac{1}{\xi_0^*}arctan\left\{\frac{(A_1 + A_2)\xi_0^*}{-(\xi_0^*)^2 + A_3 + A_4}\right\} + \frac{2k\pi}{\xi_0^*}, \quad k = 0, 1, 2, \cdots \tag{17}$$

According to Butler's lemma [28], $E^*$ remains locally asymptotically stable for $0 < \tau_1 < \tau_1^*(= \min_{k\geq 0} \tau_{1_k})$ and unstable if $\tau_1 > \tau_1^*$.

Now, we take the derivative of Equation (12) with respect to $\tau_1$ as

$$\left(\frac{d\lambda}{d\tau_1}\right)^{-1} = \frac{2\lambda + A_1 + A_2 - \tau_1 A_5 e^{-\lambda\tau_1}}{\lambda A_5 e^{-\lambda\tau_1}}$$

$$= \frac{2\lambda + A_1 + A_2}{\lambda A_5 e^{-\lambda\tau_1}} - \frac{\tau_1}{\lambda}.$$

Furthermore, because $e^{-\lambda\tau_1} = -\frac{\lambda^2+(A_1+A_2)\lambda+A_3+A_4}{A_5}$, we can derive

$$\left(\frac{d\lambda}{d\tau_1}\right)^{-1} = -\frac{2\lambda + A_1 + A_2}{\lambda(\lambda^2 + (A_1 + A_2)\lambda + A_3 + A_4)} - \frac{\tau_1}{\lambda}.$$

At $\tau_1 = \tau_1^*, \xi^* = \xi_0^*$,

$$\left(\frac{d}{d\tau_1} Re\lambda(\tau_1)\right)^{-1} = \frac{((A_1 + A_2)^2 - 2(A_3 + A_4))\xi_0^* + 2(\xi_0^*)^4}{(A_1 + A_2)^2(\xi_0^*)^4 + ((\xi_0^*)^2 - (A_3 + A_4))^2(\xi_0^*)^2}$$

$$= \frac{A_5^2 - (A_3 + A_4)^2 + (\xi_0^*)^4}{(A_1 + A_2)^2(\xi_0^*)^4 + ((\xi_0^*)^2 - (A_3 + A_4))^2(\xi_0^*)^2}$$

$$> 0.$$

Hence, the system (1) has Hopf bifurcation at $\tau_1 = \tau_1^*$.

Based on the above analysis, we have the following conclusion:

**Theorem 6.** *In the absence of $\tau_2$ and $(A_3 + A_4)^2 < A_5^2$, the coexistence equilibrium point $E^*$ is locally asymptotically stable for $0 < \tau_1 < \tau_1^*$ and unstable for $\tau_1 > \tau_1^*$. In addition, the system (1) will undergo a Hopf bifurcation at $\tau_1 = \tau_1^*$.*

Case III: $\tau_1 = 0, \tau_2 > 0$. The characteristic Equation (11) becomes

$$\lambda^2 + A_1\lambda + (A_2\lambda + A_3 + A_5)e^{-\lambda\tau_2} + A_4 = 0. \tag{18}$$

Taking $\lambda = i\bar{\xi}^*$. Then, we obtain the real and imaginary parts, respectively, as follows:

$$-(\bar{\xi}^*)^2 + (A_3 + A_5)cos\bar{\xi}^*\tau_2 + A_2\bar{\xi}^*sin\bar{\xi}^*\tau_2 + A_4 = 0, \tag{19}$$

$$A_1\bar{\xi}^* + A_2\bar{\xi}^*cos\bar{\xi}^*\tau_2 - (A_3 + A_5)sin\bar{\xi}^*\tau_2 = 0. \tag{20}$$

Squaring (19) and (20) and adding, we obtain

$$(\bar{\xi}^*)^4 + R_3(\bar{\xi}^*)^2 + R_4 = 0. \tag{21}$$

Define $R_3 = -2A_4 + A_1^2 - A_2^2, R_4 = -(A_3 + A_5)^2 + A_4^2$. By calculating, we have

$$(\bar{\xi}^*)^2 = \frac{-R_3 \pm \sqrt{R_3^2 - 4R_4}}{2}. \tag{22}$$

Furthermore, because

$$R_3 = \left(-\alpha x^* + \frac{apx^*y^*}{(ax^* + by^* + c)^2}\right)^2 + (hy^*)^2 + 2(d + hy^*)hy^* + \frac{(\mu px^*)^2(2(ax^* + c)by^* + (by^*)^2)}{(ax^* + by^* + c)^4} > 0.$$

When $R_4 > 0$, Equation (21) has no positive roots and no real $\bar{\xi}^*$ exists. So, $E^*$ is locally asymptotically stable for any $\tau_2 > 0$;

When $R_4 < 0$, Equation (22) has unique positive root $(\bar{\xi}_0^*)^2$. So, Equation (18) exists a pair of imaginary roots $\pm i\bar{\xi}_0^*$. Putting $(\bar{\xi}_0^*)^2$ in (19) and (20), we obtain

$$\tau_{2_k} = \frac{1}{\bar{\xi}_0^*}cos^{-1}\left\{\frac{((\bar{\xi}_0^*)^2 - A_4)A_2\bar{\xi}_0^* + A_1\bar{\xi}_0^*(A_3 + A_5)}{A_2(\bar{\xi}_0^*)^2 + (A_3 + A_5)^2}\right\} + \frac{2k\pi}{\bar{\xi}_0^*}, \quad k = 0, 1, 2, \cdots \tag{23}$$

According to Butler's lemma, $E^*$ is locally asymptotically stable for $0 < \tau_2 < \tau_2^*$ ($=\min_{k \geq 0} \tau_{2_k}$) and unstable if $\tau_2 > \tau_2^*$.

Now, take the derivative of Equation (18) with respect to $\tau_2$

$$\left(\frac{d\lambda}{d\tau_2}\right)^{-1} = \frac{2\lambda + A_1}{\lambda(A_2\lambda + A_3 + A_5)e^{-\lambda\tau_2}} + \frac{A_2}{\lambda(A_2\lambda + A_3 + A_5)} - \frac{\tau_2}{\lambda}.$$

Furthermore, because $e^{-\lambda\tau_2} = -\frac{\lambda^2 + A_1\lambda + A_4}{A_2\lambda + A_3 + A_5}$, we can derive

$$\left(\frac{d\lambda}{d\tau_2}\right)^{-1} = -\frac{2\lambda + A_1}{\lambda(\lambda^2 + A_1\lambda + A_4)} + \frac{A_2}{\lambda(A_2\lambda + A_3 + A_5)} - \frac{\tau_2}{\lambda}.$$

At $\tau_2 = \tau_2^*, \tilde{\xi}^* = \bar{\tilde{\xi}}_0^*$,

$$\begin{aligned}
\left(\frac{d}{d\tau_2}Re\lambda(\tau_2)\right)^{-1} &= \frac{A_1^2 + 2((\bar{\tilde{\xi}}_0^*)^2 - A_4)}{(A_1\bar{\tilde{\xi}}_0^*)^2 + ((\bar{\tilde{\xi}}_0^*)^2 - A_4)^2} - \frac{A_2^2}{(A_2\bar{\tilde{\xi}}_0^*)^2 + (A_3 + A_5)^2} \\
&= \frac{A_1^2 - A_2^2 - 2A_4 + 2(\bar{\tilde{\xi}}_0^*)^2}{(A_2\bar{\tilde{\xi}}_0^*)^2 + (A_3 + A_5)^2} \\
&= \frac{R_3 + 2(\bar{\tilde{\xi}}_0^*)^2}{A_2^2(\bar{\tilde{\xi}}_0^*)^2 + (A_3 + A_5)^2} \\
&> 0.
\end{aligned}$$

Hence, the system (1) undergoes Hopf bifurcation at $\tau_2 = \tau_2^*$.

According to the above analysis, we have the following theorem:

**Theorem 7.** *In the absence of $\tau_1$, for system (1),*

*(1) If $R_4 > 0$, the coexistence equilibrium point $E^*$ is locally asymptotically stable for any $\tau_2 > 0$;*

*(2) If $R_4 < 0$, the coexistence equilibrium point $E^*$ is locally asymptotically stable for $0 < \tau_2 < \tau_2^*$ and unstable if $\tau_2 > \tau_2^*$. In addition, the system (1) will undergo a Hopf bifurcation at $\tau_2 = \tau_2^*$.*

Case IV: $\tau_1$ is fixed in $(0, \tau_1^*), \tau_2 > 0$. Assume $\lambda = i\tilde{\xi}$, which is put in (11) and separate the real and imaginary parts as follows:

$$-\tilde{\xi}^2 + A_2\tilde{\xi}sin\tilde{\xi}\tau_2 + A_3cos\tilde{\xi}\tau_2 + A_4 + A_5(cos\tilde{\xi}\tau_1cos\tilde{\xi}\tau_2 - sin\tilde{\xi}\tau_1sin\tilde{\xi}\tau_2) = 0,$$

$$A_1\tilde{\xi} + A_2\tilde{\xi}cos\tilde{\xi}\tau_2 - A_3sin\tilde{\xi}\tau_2 - A_5(sin\tilde{\xi}\tau_1cos\tilde{\xi}\tau_2 + sin\tilde{\xi}\tau_2cos\tilde{\xi}\tau_1) = 0.$$

The above formulas can be arranged as:

$$(A_2\tilde{\xi} - A_5sin\tilde{\xi}\tau_1)sin\tilde{\xi}\tau_2 + (A_3 + A_5cos\tilde{\xi}\tau_1)cos\tilde{\xi}\tau_2 = \tilde{\xi}^2 - A_4, \tag{24}$$

$$-(A_3 + A_5cos\tilde{\xi}\tau_1)sin\tilde{\xi}\tau_2 + (A_2\tilde{\xi} - A_5sin\tilde{\xi}\tau_1)cos\tilde{\xi}\tau_2 = -A_1\tilde{\xi}. \tag{25}$$

Squaring (24) and (25) and adding to eliminate $\tau_2$, we have

$$\tilde{\xi}^4 + R_3\tilde{\xi}^2 + R_5\tilde{\xi} + R_6 = 0, \tag{26}$$

where $R_5 = 2A_2A_5sin\tilde{\xi}\tau_1, R_6 = -A_5^2 - A_3^2 + A_4^2 - 2A_3A_5cos\tilde{\xi}\tau_1 = 0$.

Define $Z(\tilde{\xi}) = \tilde{\xi}^4 + R_3\tilde{\xi}^2 + R_5\tilde{\xi} + R_6 = 0$. Then, $Z(0) = -A_5^2 - A_3^2 + A_4^2 - 2A_3A_5 = -(A_3 + A_5)^2 + A_4^2$, and $Z(\infty) = \infty$. Suppose $Z(0) < 0$, that is to say, $A_4^2 < (A_3 + A_5)^2$. Then, Equation (26) has only one positive root. Hence, there exist the roots $\pm i\tilde{\xi}^*$ in the characteristic Equation (11). From (24) and (25), we obtain

$$\tilde{\tau}_{2_k} = \frac{1}{\tilde{\xi}^*}arcsin\frac{F_1F_3 + F_2F_4}{F_1^2 + F_2^2} + \frac{2k\pi}{\tilde{\xi}^*}, \quad k = 0, 1, 2, \cdots \tag{27}$$

where $F_1 = A_2\tilde{\xi} - A_5sin\tilde{\xi}\tau_1, F_2 = A_3 + A_5cos\tilde{\xi}\tau_1, F_3 = \tilde{\xi}^2 - A_4, F_4 = -A_1\tilde{\xi}$.

According to Butler's lemma, $E^*$ is locally asymptotically stable for $0 < \tau_2 < \tilde{\tau}_2^*$ ($= \min\limits_{k \geq 0} \tilde{\tau}_{2_k}$) and unstable if $\tilde{\tau}_2^*$.

Now, take the derivative of Equation (11) with respect to $\tau_2$

$$\left(\frac{d\lambda}{d\tau_2}\right)^{-1} = \frac{2\lambda + A_1 + A_2 e^{-\lambda\tau_2} - A_5\tau_1 e^{-\lambda(\tau_1+\tau_2)}}{\lambda(A_2\lambda + A_3 + A_5 e^{-\lambda\tau_1})e^{-\lambda\tau_2}} - \frac{\tau_2}{\lambda}.$$

Furthermore, because $e^{-\lambda\tau_2} = -\frac{\lambda^2 + A_1\lambda + A_4}{A_2\lambda + A_3 + A_5 e^{-\lambda\tau_2}}$, we can derive

$$\left(\frac{d\lambda}{d\tau_2}\right)^{-1} = -\frac{2\lambda + A_1}{\lambda(\lambda^2 + A_1\lambda + A_4)} + \frac{A_2 - A_5\tau_1 e^{-\lambda\tau_1}}{\lambda(A_2\lambda + A_3 + A_5 e^{-\lambda\tau_1})} - \frac{\tau_2}{\lambda}.$$

At $\tau_2 = \tilde{\tau}_2^*, \tilde{\xi} = \tilde{\xi}^*$,

$$\left(\frac{d}{d\tau_2}Re\lambda(\tau_2)\right)^{-1} = \frac{A_1^2 + 2F_3}{F_4^2 + F_3^2} + \frac{-F_1F_5 + F_6F_2}{\tilde{\xi}^*(F_1^2 + F_2^2)^2},$$

where $F_5 = A_2 - A_5\tau_1\cos\tilde{\xi}^*\tau_1$, $F_6 = A_5\tau_1\sin\tilde{\xi}^*\tau_1$.

So, $\left(\frac{d}{d\tau_2}Re\lambda(\tau_2)\right)^{-1} > 0$ leads to $F_6F_2 - F_1F_5 > 0$. Hence, there exists Hopf bifurcation at $\tau_2 = \tilde{\tau}_2^*$ in the system (1).

Based on the above analysis, we have the following theorem:

**Theorem 8.** *Assume that $\tau_1$ is fixed in $(0, \tau_1^*)$ and $A_4^2 < (A_3 + A_5)^2$. Then, the coexistence equilibrium point $E^*$ is locally asymptotically stable for $0 < \tau_2 < \tilde{\tau}_2^*$ and unstable for $\tau_2 > \tilde{\tau}_2^*$. In addition, the system (1) will undergo a Hopf bifurcation at $\tau_2 = \tilde{\tau}_2^*$ provided $F_6F_2 - F_1F_5 > 0$, where $F_i$ are all defined in the proof.*

## 4. Stochastic Delay Model Analysis

*4.1. Existence and Uniqueness of Positive Solution*

In this subsection, we prove the unique existence of a global positive solution by means of a random comparison theorem.

**Theorem 9.** *For any given initial value $(x(0), y(0)) \in \mathbb{R}_+^2$, there is a unique solution $(x(t), y(t))$ to system (3) on $t \geq 0$. Furthermore, the solution will remain in $\mathbb{R}_+^2$ with probability 1, that is to say, $(x(t), y(t)) \in \mathbb{R}_+^2$ almost surely.*

**Proof.** Taking $m = \ln x, n = \ln y$, applying Itô's formula we have

$$dm = \left(\frac{k}{1 + fe^{n(t-\tau_1)}} - \alpha e^m - \frac{pe^n}{ae^m + be^n + c} - \frac{\sigma_1^2}{2(1 + fe^{n(t-\tau_1)})^2}\right)dt + \frac{\sigma_1}{1 + fe^{n(t-\tau_1)}}dB_1(t),$$

$$dn = \left(\frac{\mu pe^{m(t-\tau_2)}e^{n(t-\tau_2)}}{e^n(ae^{m(t-\tau_2)} + be^{n(t-\tau_2)} + c)} - d - he^n - \frac{\sigma_2^2}{2}\right) + \sigma_2 dB_2(t),$$

where $m(0) = \ln x(0)$ and $n(0) = \ln y(0)$. We notice that the coefficients of the above equation satisfy the local Lipschitz condition, so it possesses a unique local solution $(m(t), n(t))$ on $t \in [0, \tau_e)$, where $\tau_e$ is the explosion time. Hence, $x = e^m(t), y = e^n(t)$ is the unique positive local solution to (3) with initial value $x_0 > 0, y_0 > 0$. To show that the solution is global, we only need to prove $\tau_e = \infty$.

According to the first equation of (3), we have

$$dx \leq x(t)(k - \alpha x(t))dt + \sigma_1 x(t)dB_1(t).$$

Let $Y(t)$ be the unique solution of the equation

$$\begin{cases} dY(t) &=& Y(t)(k - \alpha Y(t))dt + \sigma_1 Y(t)dB_1(t), \\ Y(0) &=& x(0), \end{cases}$$

then,

$$Y(t) = \frac{e^{(k - \frac{\sigma_1^2}{2})t + \sigma_1 B_1(t)}}{\frac{1}{x(0)} + \alpha \int_0^t e^{(k - \frac{\sigma_1^2}{2})s + \sigma_1 B_1(s)} \mathrm{d}s}.$$

Hence, by the comparison theorem [29] for stochastic equations, we obtain

$$x(t) \leq Y(t). \tag{28}$$

According to the second equation of (3), we have

$$dy(t) \geq y(t)(-d - hy(t))dt + \sigma_2 y(t)dB_2(t).$$

Let $\Psi(t)$ is the unique solution of the equation

$$\begin{cases} d\Psi(t) &=& \Psi(t)(-d - h\Psi(t))dt + \sigma_2 \Psi(t)dB_2(t), \\ \Psi(0) &=& y(0), \end{cases}$$

then,

$$\Psi(t) = \frac{e^{(-d - \frac{\sigma_2^2}{2})t + \sigma_2 B_2(t)}}{\frac{1}{y(0)} + h \int_0^t e^{(-d - \frac{\sigma_2^2}{2})s + \sigma_2 B_2(s)} \mathrm{d}s}.$$

Therefore,

$$y(t) \geq \Psi(t). \tag{29}$$

On the other hand,

$$dy(t) \leq \left( \frac{\mu p Y(t - \tau_2)}{b} - dy(t) \right) dt + \sigma_2 y(t)dB_2(t).$$

Let $\Phi(t)$ be the unique solution of the equation

$$\begin{cases} d\Phi(t) &=& \left( \frac{\mu p \Phi(t - \tau_2)}{b} - dy(t) \right) dt + \sigma_2 y(t)dB_2(t), \\ \Phi(0) &=& y(0). \end{cases}$$

Let $N(t) = \frac{1}{\Phi(t)}$ with $N(0) = \frac{1}{\Phi(0)}$, by Itô's formula

$$dN(t) = \left( -\frac{\mu p Y(t - \tau_2)}{b} N^2(t) + dN(t) + \sigma_2^2 N(t) \right) dt - \sigma_2 N(t)dB_2(t)$$

$$= \left( d + \sigma_2^2 - \frac{\mu p Y(t - \tau_2)}{b} N(t) \right) N(t)dt - \sigma_2 N(t)dB_2(t),$$

then,

$$N(t) = \frac{e^{(d + \frac{\sigma_2^2}{2})t - \sigma_2 B_2(t)}}{\Phi(0) + \frac{\mu p}{b} \int_0^t Y(s - \tau_2)e^{(d + \frac{\sigma_2^2}{2})s - \sigma_2 B_2(s)} \mathrm{d}s}.$$

Hence, $\Phi(t) = \frac{1}{N(t)} = \Phi(0)e^{-(d+\frac{\sigma_2^2}{2})t+\sigma_2 B_2(t)} + \frac{\mu p}{b}e^{-(d+\frac{\sigma_2^2}{2})t+\sigma_2 B_2(t)} \int_0^t Y(s-\tau_2)$
$e^{(d+\frac{\sigma_2^2}{2})s-\sigma_2 B_2(s)}ds.$

Similarly, we can derive

$$y(t) \leq \Phi(t). \tag{30}$$

Again, from the first equation of (3), we have

$$dx(t) \geq x(t)\left(\frac{k}{1+f\Phi(t-\tau_1)} - \alpha x - \frac{p}{b}\right)dt + \frac{\sigma_1 x(t)}{1+f\Phi(t-\tau_1)}dB_1(t).$$

We can safely infer

$$x(t) \geq \frac{e^{-\frac{p}{b}t+\int_0^t\left(\frac{k}{1+f\Phi(s-\tau_1)} - \frac{\sigma_1^2}{2(1+f\Phi(s-\tau_1))^2}\right)ds+\int_0^t \frac{\sigma_1}{1+f\Phi(s-\tau_1)}dB_1(s)}}{\frac{1}{x(0)} + \alpha\int_0^t e^{-\frac{p}{b}s+\int_0^s\left(\frac{k}{1+f\Phi(\rho-\tau_1)} - \frac{\sigma_1^2}{2(1+f\Phi(\rho-\tau_1))^2}\right)d\rho+\int_0^s \frac{\sigma_1}{1+f\Phi(\rho-\tau_1)}dB_1(\rho)}ds} := \Theta(t),$$

then, we have

$$x(t) \geq \Theta(t). \tag{31}$$

Therefore, we have

$$\Theta(t) \leq x(t) \leq Y(t), \ \Psi(t) \leq y(t) \leq \Phi(t).$$

Hence, $\tau_e = \infty$, that is to say, the solution of (3) is global. This completes the proof. □

*4.2. Stochastic Ultimate Boundedness*

**Definition 1.** *Consider a stochastic differential equation $dX(t) = F(X(t))dt + S(X(t))dB(t)$. Its solution is said to be stochastic ultimate bounded if for any $\epsilon \in (0,1)$, there exists a positive $Q = Q(\epsilon)$ such that for any initial value $(x(0), y(0)) \in \mathbb{R}_+^2$, the solution satisfies*

$$\limsup_{t\to\infty} P\{|(x(t),y(t))| > Q\} < \epsilon.$$

**Theorem 10.** *For any $r \in (0,1)$ there exists a positive $Q = Q(r)$ such that the solution $(x(t), y(t))$ of the model (3) satisfies*

$$\limsup_{t\to\infty} E|(x(t),y(t))|^r \leq Q,$$

*then we can derive the solution of the model (3) is stochastic and ultimately bounded.*

**Proof.** Let $V_1(x,y) = x^r + y^r$, from Itô's formula we have

$$dV_1(x,y) = LV_1(x,y)dt + rx^r\frac{\sigma_1}{1+fy(t-\tau_1)}dB_1(t) + r\sigma_2 y^r dB_2(t),$$

where

$$LV_1(x,y) = rx^r \left( \frac{k}{1+fy(t-\tau_1)} - \alpha x - \frac{py}{ax+by+c} \right) + \frac{1}{2}r(r-1)x^r \frac{\sigma_1^2}{(1+fy(t-\tau_1))^2}$$
$$+ ry^{r-1}\left( \frac{\mu px(t-\tau_2)y(t-\tau_2)}{ax(t-\tau_2)+by(t-\tau_2)+c} - dy - hy^2 \right) + \frac{1}{2}r(r-1)\sigma_2^2 y^r$$
$$\leq \frac{krx^r}{1+fy(t-\tau_1)} - \alpha rx^{r+1} - \frac{1}{2}r(1-r)x^r \frac{\sigma_1^2}{(1+fy(t-\tau_1))^2} + x^r$$
$$+ \frac{r\mu px(t-\tau_2)y(t-\tau_2)}{ax(t-\tau_2)+by(t-\tau_2)+c}y^{r-1} - hry^{r+1} - dry^r - \frac{1}{2}r(1-r)\sigma_2^2 y^r + y^r - V_1(x,y)$$
$$\leq H_1 - V_1(x,y),$$

where $H_1$ is a positive constant. Hence, we can obtain

$$dV_1(x,y) \leq (H_1 - V_1(x,y))dt + rx^r \frac{\sigma_1}{1+fy(t-\tau_1)}dB_1(t) + r\sigma_2 y^r dB_2(t).$$

Define $V_2(x,y) = e^t V_1(x,y)$, using Itô's formula, we obtain

$$dV_2(x,y) \leq e^t H_1 dt + rx^r \frac{\sigma_1}{1+fy(t-\tau_1)}dB_1(t) + r\sigma_2 y^r dB_2(t).$$

Integrating both sides of the above inequality and taking the expectation, then

$$e^t EV_1(x,y) \leq (e^t - 1)H_1 + V_1(x(0), y(0)).$$

So,

$$EV_1(x,y) \leq (1-e^{-t})H_1 + e^{-t}V_1(x(0), y(0)).$$

Hence, we can obtain

$$\limsup_{t\to\infty} EV_1(x,y) \leq H_1.$$

Because

$$|(x(t), y(t))|^r = (x^2(t) + y^2(t))^{\frac{r}{2}} \leq 2^{\frac{r}{2}} \max\{x^r, y^r\} \leq 2^{\frac{r}{2}} V_1(x,y),$$

we can derive

$$\limsup_{t\to\infty} E|(x(t), y(t))|^r \leq 2^{\frac{r}{2}} \limsup_{t\to\infty} EV_1(x,y) \leq 2^{\frac{r}{2}} H_1 := Q.$$

For any $\epsilon > 0$, let $H_1 = (\frac{H}{\epsilon})^{\frac{1}{r}}$. According to the Markov inequality, we have

$$P\{|(x(t), y(t))| > H_1\} < \frac{E|(x(t), y(t))|^r}{H_1^r} < \epsilon.$$

The proof is complete.  □

## 5. Numerical Simulations

In this section, as a general method in stochastic differential equations, we validate our mathematical findings by performing numerical simulations using Milstein's high-order method [30] and MATLAB 2019a, that is, as proven in the previous sections. We select the following parameters:

$$f = 1.5, p = 4, a = 4.8, b = 5, c = 2.1, \mu = 1.2, d = 0.1, h = 0.01, k = 0.9, \alpha = 0.6, \quad (32)$$

and take the initial data $x(0) = 0.5, y(0) = 0.5$. For the parameters (32), the system (1) has three equilibrium points, which are the trivial equilibrium point $E_0(0,0)$, the free equilibrium point $E_1(1.5, 0)$, and the coexistence equilibrium point $E^*(x^*, y^*) = (0.1219, 0.5702)$. By Theorem (3), $E_0$ is always unstable. Furthermore, the existence of $E^*$ confirms that $E_2$ is unstable for any $\tau_1 \geq 0$ and $\tau_2 \geq 0$.

Case I: Assume $\tau_1 = 0$ and $\tau_2 = 0$ and keep other parameters the same as (32). By simple calculation, $\frac{apy^*}{\alpha(ax^* + by^* + c)^2} - 1 = -0.4047 < 0$. Then, the condition of Theorem 5 is satisfied, which implies that $E^*$ is locally asymptotically stable, as shown in Figure 1.

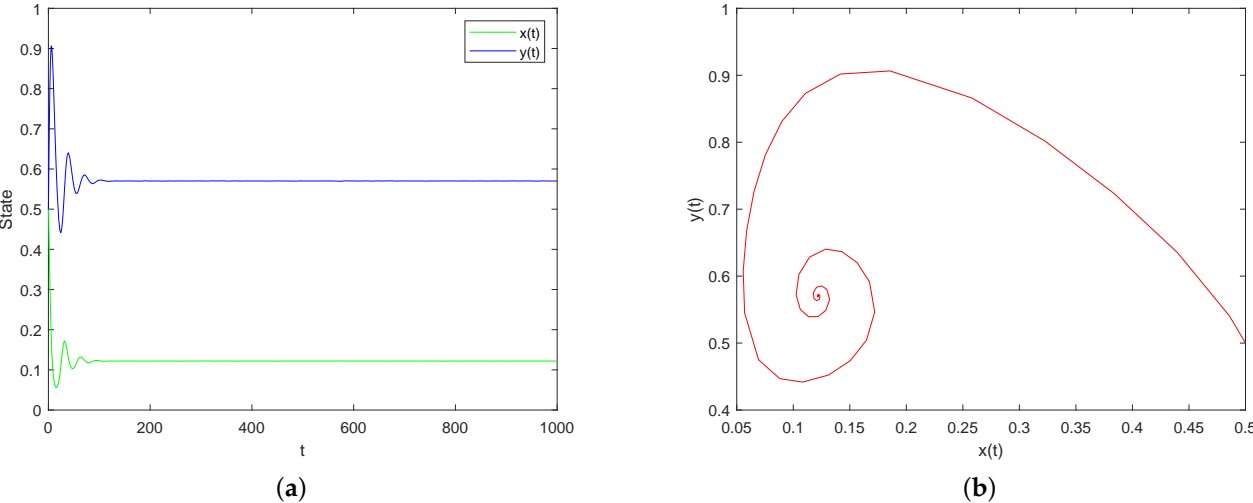

(a)　　　　　　　　　　　　　　　　　　　(b)

**Figure 1.** (a) Time series and (b) phase portrait of $E^*$ for system (1) when $\tau_1 = 0$ and $\tau_2 = 0$. Other parameters are the same as in (32).

Case II: Assume $\tau_2 = 0$ and slowly increase the value of $\tau_1$. Leave the other parameters remain (32) unchanged. By virtue of Theorem 6, we can obtain the critical value of $\tau_1 = \tau_1^* = 4.5923$. When $\tau_1 = 3.5 < \tau_1^*$, the system (1) is locally asymptotically stable (as shown in Figure 2) and unstable for $\tau_1 = 5 > \tau_1^*$. Furthermore, the occurrence of oscillation behavior and limit cycle is illustrated by Figure 3. Furthermore, the system (1) undergoes a Hopf bifurcation at $\tau_1 = \tau_1^*$ as shown in Figure 4.

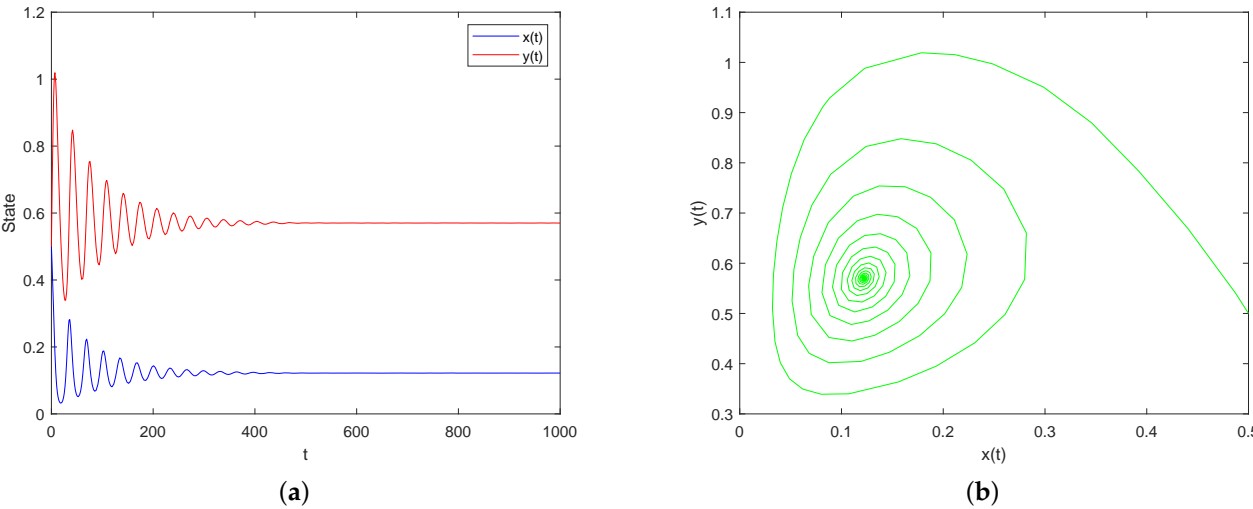

(a)　　　　　　　　　　　　　　　　　　　(b)

**Figure 2.** (a) Time series and (b) phase portrait of $E^*$ for system (1) when $\tau_1 = 3.5$ and $\tau_2 = 0$. Other parameters are the same as in (32).

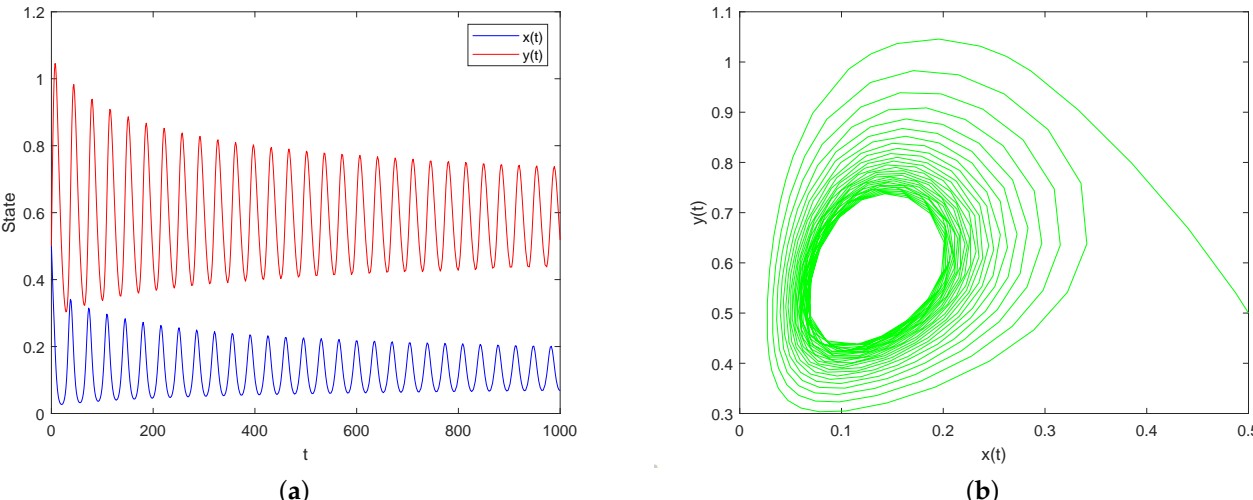

**Figure 3.** (**a**) Time series and (**b**) phase portrait of $E^*$ for system (1) when $\tau_1 = 5$ and $\tau_2 = 0$. Other parameters are same as in (32).

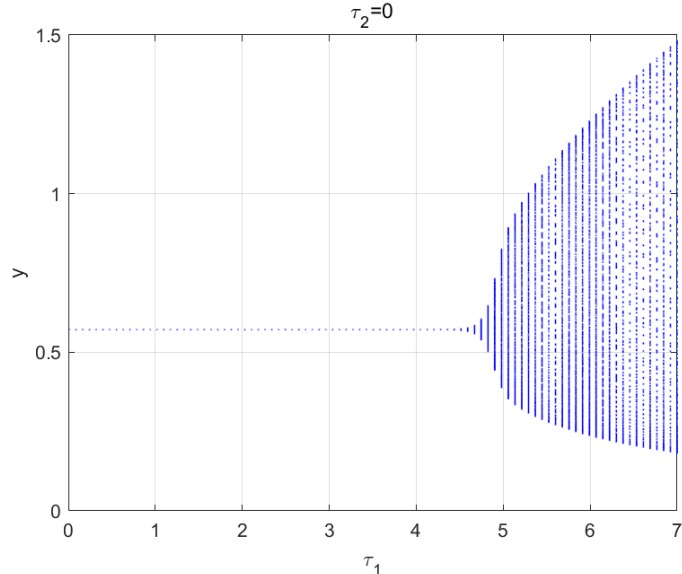

**Figure 4.** The Hopf bifurcation diagram of (1) for $\tau_1$ and $\tau_2 = 0$.

Case III: Assume $\tau_1 = 0$ and gradually increase the value of $\tau_2$, keeping other parameters remaining in (32) unchanged. After numerical calculation $R_4 = -0.0015 < 0$, according to Theorem 7, we can obtain the critical value of $\tau_2 = \tau_2^* = 2.3941$. When $\tau_2 = 1.5 < \tau_2^*$, the system (1) is locally asymptotically stable as shown in Figure 5 and unstable for $\tau_2 = 2.5 > \tau_2^*$, which is illustrated by Figure 6. Furthermore, the system (1) experiences a Hopf bifurcation at $\tau_2 = \tau_2^*$ (as shown in Figure 7).

Case IV: We take a fixed $\tau_1 = 3.5 \in (0, \tau_1^*) = (0, 4.5923)$ and vary the parameter $\tau_2$ while keeping other parameters the same as in (32). From Theorem 8, we can see that the critical value of $\tau_2 = \tilde{\tau}_2^* = 0.4830$. When $\tau_2 = 0.2 < \tilde{\tau}_2^*$, the system (1) is locally asymptotically stable as shown in Figure 8 and unstable for $\tau_2 = 0.5 > \tilde{\tau}_2^*$, which is illustrated by Figure 9. Figure 10 reveals that a Hopf bifurcation occurs at $\tau_2 = \tilde{\tau}_2^*$.

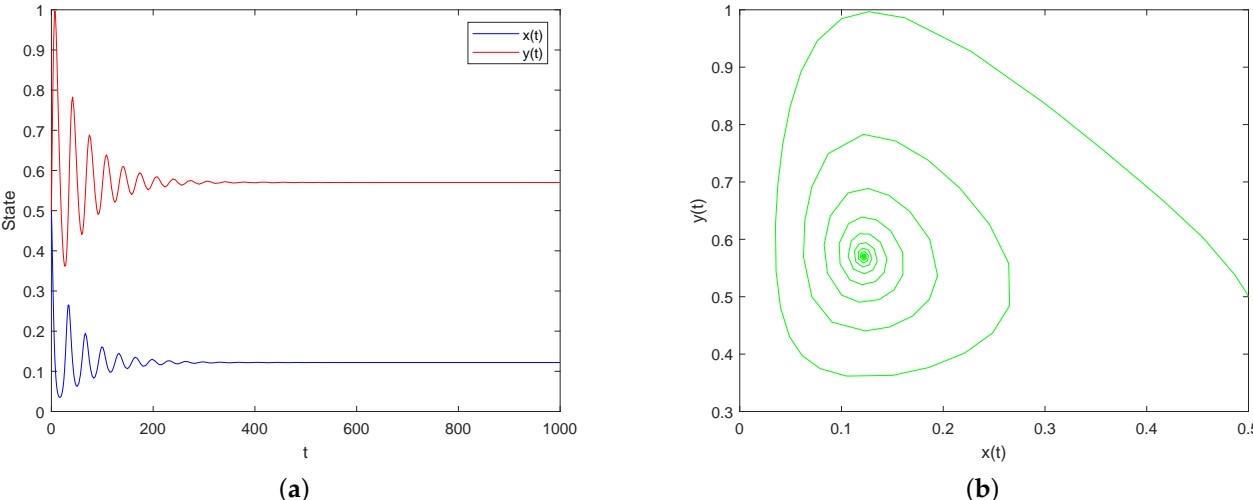

**Figure 5.** (**a**) Time series and (**b**) phase portrait of $E^*$ for system (1) when $\tau_2 = 1.5$ and $\tau_1 = 0$. Other parameters are the same as in (32).

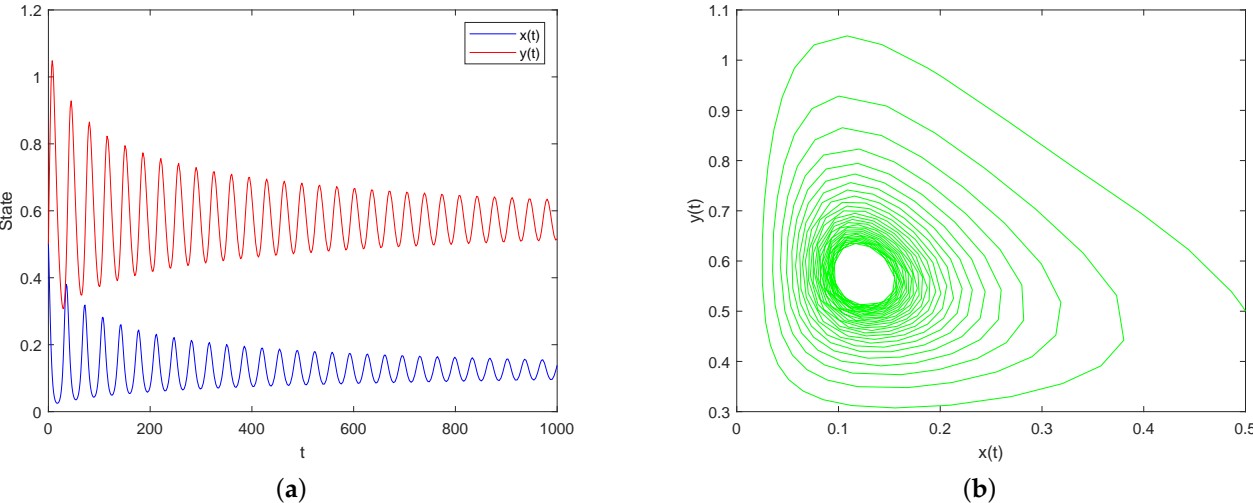

**Figure 6.** (**a**) Time series and (**b**) phase portrait of $E^*$ for system (1) when $\tau_2 = 2.5$ and $\tau_1 = 0$. Other parameters are the same as in (32).

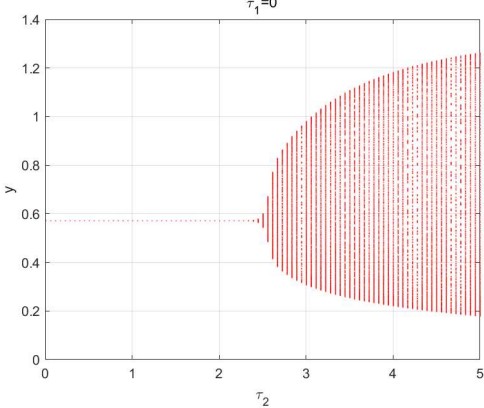

**Figure 7.** The Hopf bifurcation diagram of (1) for $\tau_2$ and $\tau_1 = 0$.

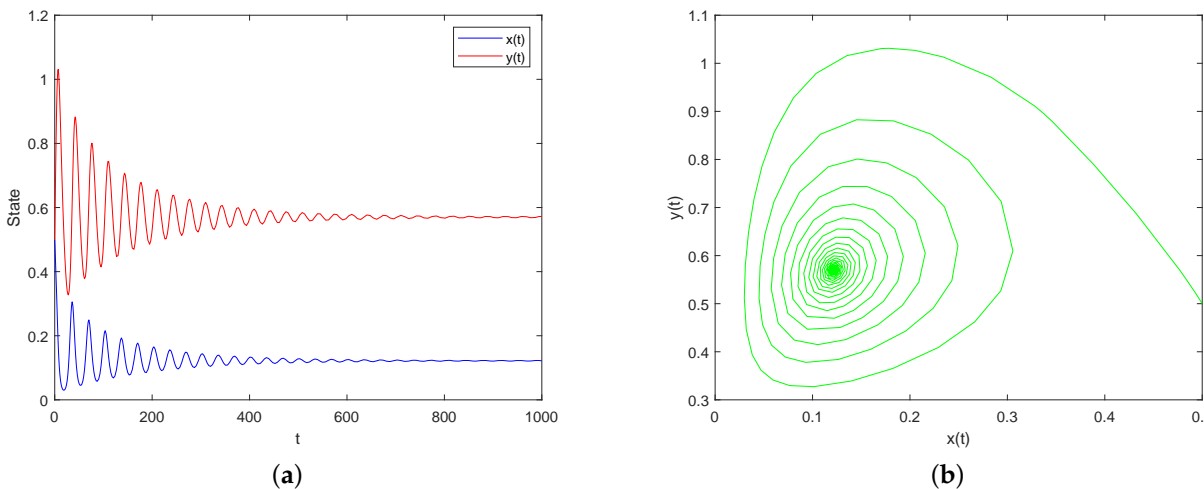

**Figure 8.** (**a**) Time series and (**b**) phase portrait of $E^*$ for system (1) when $\tau_1 = 3.5$ and $\tau_2 = 0.2$. Other parameters are the same as in (32).

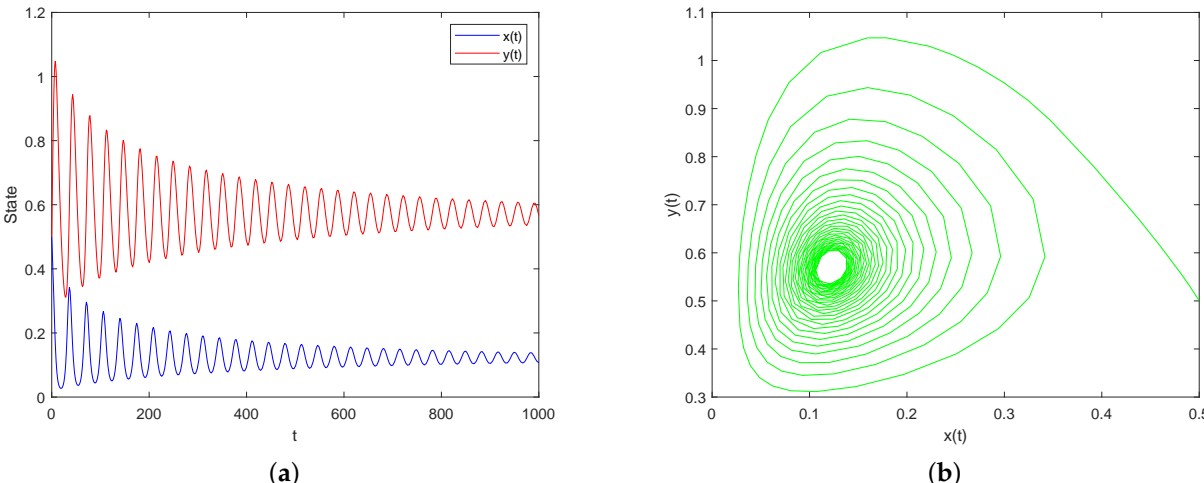

**Figure 9.** (**a**) Time series and (**b**) phase portrait of $E^*$ for system (1) when $\tau_1 = 3.5$ and $\tau_2 = 0.5$. Other parameters are the same as in (32).

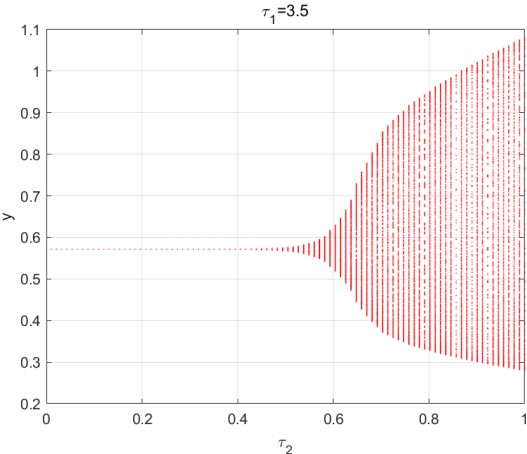

**Figure 10.** The Hopf bifurcation diagram of (1) for $\tau_2$ and $\tau_1 = 3.5$.

For stochastic delay systems (3), according to Theorem 10, it is random and ultimately bounded. Figure 11 confirms our results.

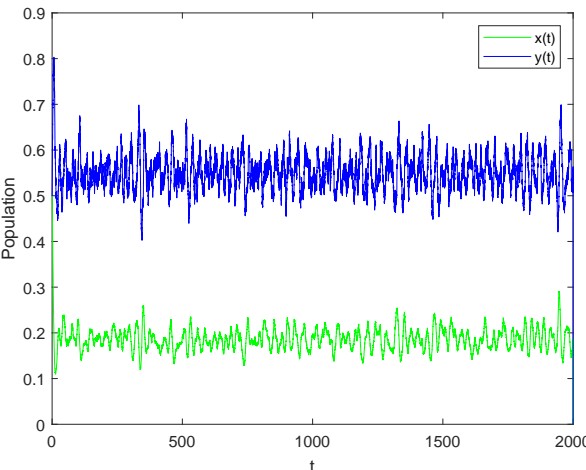

**Figure 11.** Dynamical behaviors of (3). The parameters are taken as (32) and $\sigma_1 = 0.03, \sigma_2 = 0.03$.

## 6. Conclusions

In this paper, we mainly investigate a predator–prey model with Beddington–DeAngelis functional response and fear by predator. At the same time, because predator–prey interactions do not occur immediately, we introduced the fear delay and the pregnancy delay to make the model more natural.

First, we prove the positivity and boundedness of system (1). Then, we discuss the existence criterion and local stability of equilibrium point, and find that the existence of $E^*$ ensures that $E_1$ is unstable. Finally, the existence of a Hopf bifurcation with the fear delay $\tau_1$ and the pregnancy delay $\tau_2$ as the bifurcation parameter is studied, and the critical values of the bifurcation parameters are derived in several possible cases. We find the relationship between local stability and critical bifurcation value of the system (1). When the delay is less than the critical bifurcation value, both prey and predator oscillate periodically for a finite time and then reach equilibrium. When the delay exceeds the critical bifurcation value, Hopf bifurcation occurs in system (1), and periodic oscillation and limit cycle are generated. At this point, the system (1) switches from a stable state to an unstable state. Numerical simulations confirm our theoretical findings. Furthermore, for stochastic delay system (3), we study the unique existence of the global positive solution and explore the stochastic ultimate boundedness.

In future studies, we can generalize the model (1) to a multi-population model and introduce prey refuge to investigate its impact on the stability and persistence of population dynamics. In the meantime, it may be possible to study prey refuge as a Hopf bifurcation parameter. We leave these interesting questions for further study.

**Author Contributions:** Writing—original draft preparation, M.C.; writing—review and editing, Y.S., R.X. and J.Z.; visualization, M.C.; supervision, Y.S. and R.X.; funding acquisition: Y.S. and J.Z. All authors have read and agreed to the published version of the manuscript.

**Funding:** This work is supported by Science and Technology Program of Inner Mongolia Autonomous Region (2022YFHH0063, 2022YFHH0017) and National Natural Science Foundation of China (11861027, 12161062).

**Data Availability Statement:** Not applicable.

**Acknowledgments:** The authors are very grateful to the anonymous referees and the editors for their careful reading and valuable comments, which have helped improve the presentation of this work significantly.

**Conflicts of Interest:** The authors declare no conflict of interest.

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
