# Peer review of "Dynamic Behavior of a Predator–Prey Model with Double Delays and Beddington–DeAngelis Functional Response"

_axioms, doi:10.3390/axioms12010073_

Round 1
Reviewer 1 Report
Report on the manuscript: Dynamic behavior of a predator-prey model with double delays and Beddington-DeAngelis functional response
Authors: Minjuan Cui , Yuanfu Shao1, Renxiu Xue, Jinxing Zhao
This paper deals with a delayed predator-prey models performing both theoretical an numerical study. Bifurcation is analyzed taking the time as parameter. The paper is interesting and can be accepted for publication in Axioms, although I would like to make the following comments to the authors.
Comments:
1. The authors make use of the software MATLAB as claimed, but which is the numerical method applied to solve the system?
2. Related to the previous point, please add some description on the numerical scheme, and why it has ben chosen.
2. What about the size of the time step in the numerical simulations? Are there numerical stability restrictions?
3. This kind of models are important in biology. Could the authors suggest some example in this context?
4. Please update the bibliography, since it is rather old.
Reviewer 2 Report
See the attached file
